# StylizedFacePoint: Facial Landmark Detection for Stylized Characters

## ABSTRACT

Facial landmark detection forms the foundation for numerous face-related tasks. Recently, this field has gained substantial attention and made significant advancements. Nonetheless, detecting facial landmarks for stylized characters still remains a challenge. Existing approaches, which are mostly trained on real-human face datasets, struggle to perform well due to the structural variations between real and stylized characters. Additionally, a comprehensive dataset for analyzing stylized characters' facial features is lacking. This study proposes a novel dataset, the Facial Landmark Dataset for Stylized Characters (FLSC), which contains 2674 images and 4086 faces selected from 16 cartoon video clips, together with 98 landmarks per image, labeled by professionals. Besides, we propose StylizedFacePoint: a deep-learning-based method for stylized facial landmark detection that outperforms the existing approaches. This method has also proven to work well for characters with styles outside the training domain. Moreover, we outline two primary types of applications for our dataset and method. For each, we provide a detailed illustrative example.

## CCS CONCEPTS

• **Computing methodologies → Interest point and salient region detections**; *Image processing*.

## KEYWORDS

facial landmark detection, stylized face, neural networks, landmark-related application

## 1 INTRODUCTION

Facial landmark detection refers to the process of detecting a series of predetermined landmarks on the face. The positions of these landmarks provide crucial information about features such as the approximate facial structure and expression. Consequently, facial landmark detection serves as a fundamental task in numerous face-related applications[4, 7, 19].

Currently, there are numerous well-performing methods in the field of facial landmark detection[14, 17, 31]. However, the detection of landmarks specifically for stylized characters has received little exploration and continues to face several challenges. To begin with, significant variances in facial structure, the positioning of facial features, and proportions are evident between stylized characters

and human faces. Additionally, each stylized character possesses its own distinct artistic style, adding further intricacy. Of utmost importance is the current absence of a facial landmark dataset tailored specifically to stylized characters.

To address these problems and fill the absence in this area, we amassed a large collection of images from cartoon movies and created a Facial Landmark dataset for Stylized Characters (FLSC). This dataset comprises 2674 images and 4086 annotated stylized faces. To maintain consistency with previous research, we annotated 98 landmarks for each face. This dataset can be a valuable resource for further face-related research.

To better extract the facial landmarks for stylized characters, we propose StylizedFacePoint: a novel method designed to improve the precision of stylized facial landmark detection. This method employs a stacked hourglass network[40] as its primary architecture. After each level of the hourglass network, we incorporate regression for offset heatmaps and neighbor heatmaps on $x$-axis and $y$-axis, respectively[17]. The heatmaps generated by the current stage of hourglass network are concatenated with the input features after merging, serving as the input for the subsequent stage of the network. This method helps to capture the complex structure of stylized faces and achieve higher accuracy compared to existing methods.

Finally, we demonstrate several applications of stylized characters' facial landmark detection. The first category is to utilize landmarks as a bridge variable, and we give a detailed example of how to connect audio and controller values using our proposed dataset and method. The second category is to transfer the domain of some current tasks which rely on facial landmarks, and we also exemplify this category of applications by introducing the case of 3D face reconstruction.

In summary, our main contributions can be summarized as follows:

- We create a facial landmark dataset for stylized characters, which could be widely utilized in future research.
- We propose an approach named StylizedFacePoint for landmark detection. It achieves higher accuracy compared to other methods.
- We propose several applications based on landmark detection for stylized characters.

## 2 RELATED WORK

### 2.1 Human Facial Landmark Datasets

Here we present a brief introduction to existing human facial landmark datasets:

- **300 Faces in-the-Wild (300W):** consists of 3148 train samples and 689 test samples. Test samples have 300 indoor and 300 outdoor, in-the-wild images. It covers a large variation of identity, expression, illumination conditions, pose, occlusion, and face size.

*ACM MM, 2024, Melbourne, Australia*
© 2024 Copyright held by the owner/author(s). Publication rights licensed to ACM.
ACM ISBN 978-x-xxxx-xxxx-x/YY/MM
https://doi.org/10.1145/nnnnnnn.nnnnnnn

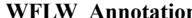

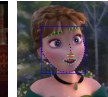
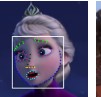
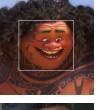
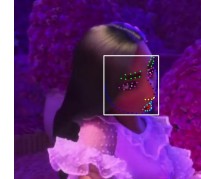
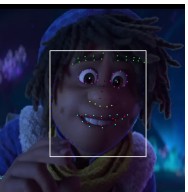
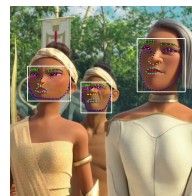

| **WFLW Annotation** | **Normal Case** | **Large Pose** | **Low-light** | **Multiple Faces** |

Figure 1: *Left:* WFLW annotation  *Right:* Samples of labeled images. Our dataset contains samples with large poses, multiple faces, or low-light conditions. Images ©Disney Enterprises, Inc.

| Dataset Name | Landmark Number | Train & Test Samples | Data Domain |
|---|---|---|---|
| 300W [28] | 68 | 3148 + 689 | human face |
| AFLW [20] | 21 | 20000 + 4386 | human face |
| COFW [3] | 29 | 1345 +507 | human face |
| WFLW [38] | 98 | 7500 + 2500 | human face |
| jha,et al. [16] | 15 | 600+150 | 2D cartoon face |
| Stricker,et al.[30] | 60 | 1157+289 | 2D manga face |
| Artistic-Faces [41] | 68 | 128+32 | 2D artistic face |
| Sindel et al. [29] | 68 | 2361+80 | 2D artistic face |
| FLSC | 98 | 3274 + 812 | 3D cartoon face |

Table 1: Comparison between current facial landmark datasets and our FLSC dataset.

- **Annotated Facial Landmarks in-the-Wild (AFLW)** have 24386 annotated images. The dataset contains a wide variety of natural face positions in addition to frontal and near-frontal faces.
- **Caltech Occluded Faces in-the-Wild (COFW)** are more difficult in terms of occlusion and position as it aims to show faces in realistic settings. The average occlusion rate for faces in this dataset is 28%, with variable degrees of occlusion, and types of occlusion vary a lot.
- **Wider Facial Landmarks in-the-Wild (WFLW)** contains 10000 face photos that were captured in many circumstances. Besides landmarks, it also has comprehensive attribute annotations. i.e., occlusion, pose, make-up, illumination, blur, and expression for analysis.

There are also some stylized landmark datasets currently available. However, these datasets suffer from a limitation in terms of the quantity of images. In contrast, our dataset boasts a larger image corpus and encompasses a wider variety of styles. We will provide a brief introduction to them in Section 2.3. A comparison between these datasets and our dataset (FLSC) is presented in Table 1.

## 2.2 Regular Facial Landmark Detection

In early computer vision tasks, traditional methods like NLoG and DoG [22] were used for landmark detection. Belhumeur et al. [1] proposed non-parametric global detectors combined with local detectors for joint landmark prediction. Markuš et al. [24] used binary decision trees to locate landmarks efficiently. With the development

of neural networks, landmark detection has benefited from deep learning. Coordinate Based Regression (CBR) and Heatmap Based Regression (HBR) are now the most popular methods, which tend to predict coordinates and heatmaps respectively.

*Coordinate Based Regression:* The CBR method requires fewer parameters but lacks precision. Researchers enhance it by cascading CNN modules for global-to-local transition [23, 26, 31, 32]. Another approach is incorporating an auxiliary network like PFLD [14]. Novel loss functions include ACR loss [11], using a logarithmic function, and Wing loss [13], adapting shape and gradient adjustments based on error magnitude. Other techniques like AnchorFace [39] employ anchors and a separation and aggregation solution strategy to tackle pose variations, and RetinaFace [6] detects minute faces in dense crowds while maintaining landmark regression precision. The ATF [21] model is trained using various datasets for landmark detection. It integrates a feature extraction network with multiple landmark regression networks, thereby enhancing the precision of the feature extraction module.

*Heatmap Based Regression:* The HBR method achieves higher landmark accuracy but with a longer running time and a larger parameter size. Optimizations include generative and adversarial networks [2, 8, 27, 45] and semi-supervised learning techniques for pseudo-label generation [9, 10, 17]. Efficient networks such as the stacked hourglass network[40], U-Net[5], and HRNet[35] are commonly utilized for generating high-quality heatmaps.Paper [12] uses knowledge distillation to compress and transfer the original large-parameter network model. Several studies have employed cascaded CNNs in the CBR method, thereby further enhancing the efficacy of the HBR method [34]. Ada loss [33] and Adaptive Wing loss [37] adjust the loss function for heatmap characteristics. Moreover, STAR loss [44] assesses the anisotropism of the predicted distribution to capture the semantic ambiguity inherent in heatmap regression.

## 2.3 Stylized Facial Landmark Detection

Compared to facial landmark detection for real human faces, research on stylized faces is relatively scarce. Agarwal et al. [16] manually label landmarks for 750 cartoon images and combined them with 2000 facial images to create a new training set. Stricker et al. [30] use a deep alignment network on the Manga109 dataset, and modify the landmarks definition based on the facial characteristics of anime characters. Jordan et al. [41] employed a three-step

training process in their own dataset, which includes original images from the art face dataset and art-style images generated by applying style transfer to facial images. Sindel et al. [29] performed landmark detection on art-style faces. Besides style transferring, they also use geometric transformations to expand their dataset. Their network architecture used global and local networks with cascaded ResNet modules for landmark detection. Huo et al. [15] also performed a 17-landmark detection on web caricature.

However, the two popular stylized landmarks detectors [29, 41] currently excel primarily on artistic faces, showing poorer performance on images with other styles. Moreover, despite training on our new dataset, the existing landmark detection models still demonstrate certain biases. Consequently, we propose StylizedFacePoint, a novel landmark detection model based on heatmap regression. We will provide a detailed introduction to our method in Section 4.

## 2.4 Applications of Landmark Detection

The accurate localization of facial landmarks plays a pivotal role in extracting crucial information regarding facial features, expressions, and various other facial attributes. Consequently, a substantial number of studies in the domain of facial analysis are centered around the detection and localization of facial landmarks. As an illustration, the Deformable Style Transfer [19] method necessitates the precise alignment of specific facial landmarks across facial images and stylized facial images. By accurately detecting facial landmarks in both images, enhanced key point matching outcomes can be achieved, thereby yielding superior style transfer effects. The incorporation of facial landmark information is also essential in numerous 3D facial reconstruction studies. As an instance, Deng et al. [7] employed 2D facial images to regress the positions of 3D landmarks, employing the distance from ground truth as the supervised loss for network training. Similarly, Cai et al. [4] employed 68 facial landmarks extracted from 2D caricatures to regress the shape and orientation of the 3D mesh. We also demonstrate several applications of stylized characters' facial landmark detection using our StylizedFacePoint landmark detector in Section 6.

## 3 DATASET DESIGN

To the best of our current knowledge, a dataset focused on 3D stylized characters' facial landmark detection is notably absent. Therefore, we introduce the Facial Landmark Dataset for Stylized Characters (FLSC) dataset to address this void.

## Image Selection

Our approach begins with the careful selection of 16 video clips from famous cartoon movies, containing more than 50 characters. These clips mostly consist of characters engaged in singing activities, which contain rich facial movements. For each video, we systematically capture one image every 5 frames. After that, we ensure a diverse representation of facial expressions. From this collection of images, we manually select those with clear facial features and exclude images with high similarity. These steps ensure a proper and diverse representation of facial expressions. Cases of multi-faces and non-front-face are also included in our dataset. Detailed information and comparisons with other landmark datasets are elaborated in Table 1.

## Landmarks Model

The selection of a suitable landmarks model is crucial for accurate facial landmark detection and following works. From various alternatives, we choose the landmark model proposed in the WFLW dataset, notable for its comprehensive 98-point representation per face. This choice provides a relatively detailed depiction of facial attributes, enhancing the dataset's utility.

## Annotation Process

Expert data annotators were engaged to label the images in our dataset. Subsequently, We manually validate the accuracy and consistency of the annotations. Apart from landmark points, our dataset also includes bounding boxes and facial expressions. These expressions are categorized into seven distinct emotions: Neutral, Sad, Surprise, Happy, Angry, Disgust, and Afraid.

By meticulously designing the FLSC dataset and employing robust annotation processes, we aim to contribute a valuable resource for stylized characters' facial landmark detection and analysis.

## 4 METHOD

In this section, we will introduce a new method: StylizedFacePoint, for landmark detection. The Hourglasses Network[42], serving as a foundational model, is presently prevalent in facial landmark regression tasks. Our approach leverages a stacked Hourglass Network [40] as its backbone. Additionally, we integrate offset regression prediction and neighbor regression prediction into the heatmap regression of landmarks. The pipeline of StylizedFacePoint is illustrated in Figure 2.

A facial image ($3 \times W \times H$) undergoes processing within the feature extraction module, comprising CNNs and residual networks. The extracted features ($C \times W' \times H'$) are then forwarded to the first-stage hourglass network along with its corresponding residual block. The feature output from the $(i-1)$-th stage hourglass network is connected with the input feature of the $(i-1)$-th stage, serving jointly as the input for the $i$-th stage. This iterative stacking of networks enables progressive extraction of facial features, enhancing the model's capability. Subsequently, after traversing through $S$ stages of networks, precise landmark location information is determined. Each hourglass network generates $N$ heatmaps, where $N$ denotes the number of pre-defined facial landmarks. These normalized heatmap depict the probability distribution of predicted facial landmarks. Finally, the Soft-Argmax function[25] is employed to infer the landmark coordinates $p$ from a single heatmap $h$:

$$p = SoftArgmax(h) = \sum_i h_i p_i \quad (1)$$

Where $p_i$ is the probability of the predicted landmark located in $y_i \in \mathbb{R}^2$.

By employing the stacked hourglass network, we obtain $N$ heatmaps of facial landmarks. The high-resolution image ($W \times H$) undergoes processing within the feature extraction network and the stacked hourglass network, yielding low-resolution heatmaps ($W' \times H'$).

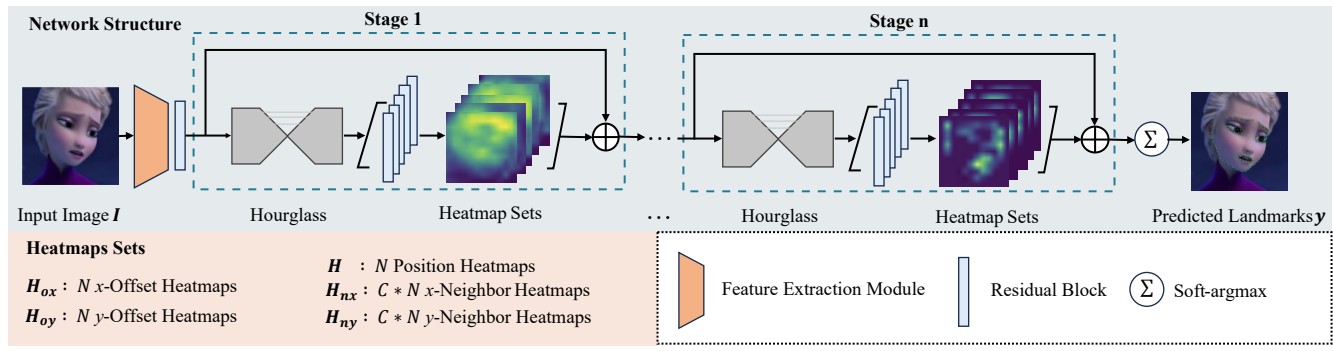

**Figure 2: The architecture of StylizedFacePoint. We use a stacked Hourglasses Network of 4 stages. Following each stage of the hourglass network, the offset regression module and neighbor regression module are applied (Input and ouput images ©Disney Enterprises, Inc.).**

To enhance the precision of heatmap regression, we introduce two offset regression modules separately on $x$-axis and $y$-axis. Additionally, considering the local positional correlation among landmarks, we integrate neighbor regression modules on $x$-axis and $y$-axis as supervision for training.

After each stage of the hourglass network outputs features, they traverse through five residual modules to generate five types of heatmaps, comprising: 1. $N$ landmark position heatmaps; 2. $N$ prediction offset heatmaps on $x$-axis; 3. $N$ prediction offset heatmaps on $y$-axis; 4. $B * N$ neighbor prediction offset heatmaps on $x$-axis, where $B$ represents the number of neighbors of the current landmark; 5. $B * N$ neighbor prediction offset heatmaps on $y$-axis.

The heatmap regression loss $L_H$ can be defined as:

$$L_H = \frac{1}{NWH} \sum_{i=1}^{N} \sum_{j=1}^{W} \sum_{k=1}^{H} (h_{ijk} - h_{ijk}^*)^2 \tag{2}$$

Here, $N$ represents the number of landmarks. $W$ and $H$ represent the width and height of the output heatmap. $h_{ijk}$ and $h_{ijk}^*$ represent the predicted and ground truth probability score value within the corresponding region of the heatmap, respectively.

Due to the potential decrease in accuracy caused by smaller-sized heatmaps, we introduce the offset regression module. Each output offset map represents the offset between the precise position of the landmark and the approximate position obtained from the heatmap. The offset regression loss $L_O$ can be defined as:

$$L_O = \frac{1}{2N} \sum_{h_{ijk}^*=1} \sum_{p=1}^{2} |o_{ijkp} - o_{ijkp}^*| \tag{3}$$

Here, $o_{ijkp}$ and $o_{ijkp}^*$ represent the predicted and ground truth offset value within the corresponding region of the heatmap, respectively.

To further enhance the accuracy of heatmap regression results, we introduce the neighbor regression module. During training, the neighbor regression module is utilized to predict the offset maps for $B$ neighbor landmarks of the current landmark. The neighbor regression loss $L_N$ can be defined as:

$$L_N = \frac{1}{2NB} \sum_{h_{ijk}^*=1} \sum_{p=1}^{2} \sum_{b=1}^{B} |n_{ijkpb} - n_{ijkpb}^*| \tag{4}$$

Here, $n_{ijkpb}$ and $n_{ijkpb}^*$ represent the predicted and ground truth offset distance between the current landmark and its neighbor landmarks, respectively.

The total loss function of multi-branch heatmap regression module $L_{total}$ consists of three components: heatmap regression loss $L_H$, offset regression loss $L_O$, and neighbor regression loss $L_N$:

$$L_{total} = \alpha_H L_H + \alpha_O L_O + \alpha_N L_N \tag{5}$$

Where $\alpha_H$, $\alpha_O$, $\alpha_N$ denote the balancing coefficients of three branches, respectively.

## 5 EXPERIMENTS

The experiments are conducted on our proposed dataset and the WFLW dataset. We compare the results of two training methods. Firstly, we train the model solely on FLSC dataset, referred to as the retraining model. Secondly, we train the model on the human dataset and subsequently fine- tune the pre-trained model using FLSC dataset, referred to as the fine-tuning model. We conduct tests for both methods on our dataset.

### 5.1 Evaluation Metric

For evaluation purposes, we utilize the normalized mean error (NME). The NME is calculated as the average Euclidean distance between the predicted locations of facial landmarks $p_{ij}$ and their corresponding ground-truth annotations $p_{ij}^*$.

$$NME = \frac{1}{NL} \sum_{i=1}^{N} \sum_{j=1}^{L} \frac{|p_{ij} - p_{ij}^*|}{d} \tag{6}$$

where $N$ is the total number of images in the testing set, $L$ is the number of landmarks, and $d$ is the chosen normalization factor. In our study, we employ the "inter-ocular" normalization factor, which corresponds to the distance between the outer corners of the two eyes[4, 19].

**Style in Dataset**                                    **Other Style**

**Trained on WFLW dataset**

**Trained on FLSC dataset**

Figure 3: Comparison between the results of models trained on different datasets. Our dataset sufficiently provides landmark-related information of the stylized face. Images ©Disney Enterprises, Inc.

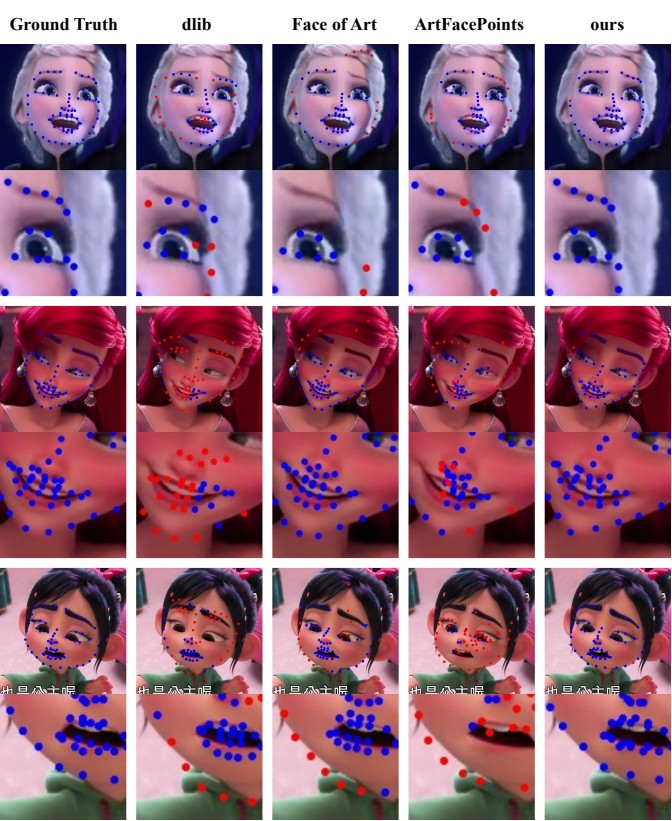

Figure 4: Comparison between three existing stylized landmark detectors and our detector. *(Red dots indicate results with significant deviations(8px) from the ground truth, and blue dots indicate other points)* Images ©Disney Enterprises, Inc.

We also export FR and AUC as evaluation metrics. The thresholds for these two metrics are both set to 0.1.

## 5.2 Implementation Details

In our method, the initial images are cropped and resized to a fixed size, i.e. 256×256, according to the annotated bounding boxes. Prior to training, we calculate the average facial landmark coordinates based on all images in the dataset. This allows us to select the $B$ closest landmarks as neighbors for each landmark. The heatmap output of the regression network is set to $16 \times 16$. In the neighbor regression module, the value of $B$, representing the number of neighbor landmarks, is set to 10. During training, the total number of epochs is set to 60. The initial learning rate is set to 1e-4, with learning rate decay applied at epochs 30 and 50. The batch size is set to 16. The values of $\alpha_H$, $\alpha_O$, and $\alpha_N$ are set to 1, 0.1, and 0.1, respectively.

## 5.3 Quantitative Evaluation

We compare our approach with open-sourced state-of-the-art methods[1]. We trained these models on the FLSC training set. Additionally, we trained separate models using retraining method and fine-tuning method. The performance comparisons of these methods on FLSC test set are shown in Table 2.

It is shown that our method achieve a slightly better result compared to existing landmark detection methods when applied to stylized faces. Moreover, within the current dataset, the retraining approach produces more precise results compared to the fine-tuning method.

We also conducted a comparison between our model and existing stylized facial landmark detection models. The performance comparisons of these models on FLSC dataset are presented in Table 3. It indicate that our model outperforms the currently available models in detecting landmarks on stylized faces.

## 5.4 Visualized Evaluation

To demonstrate the effectiveness of our approach and the necessity of our dataset, we conduct three visualized comparisons.

---

[1]Some methods, including PFLD [14] and SPICA [26], use additional pose data, therefore can't be evaluated on our dataset.

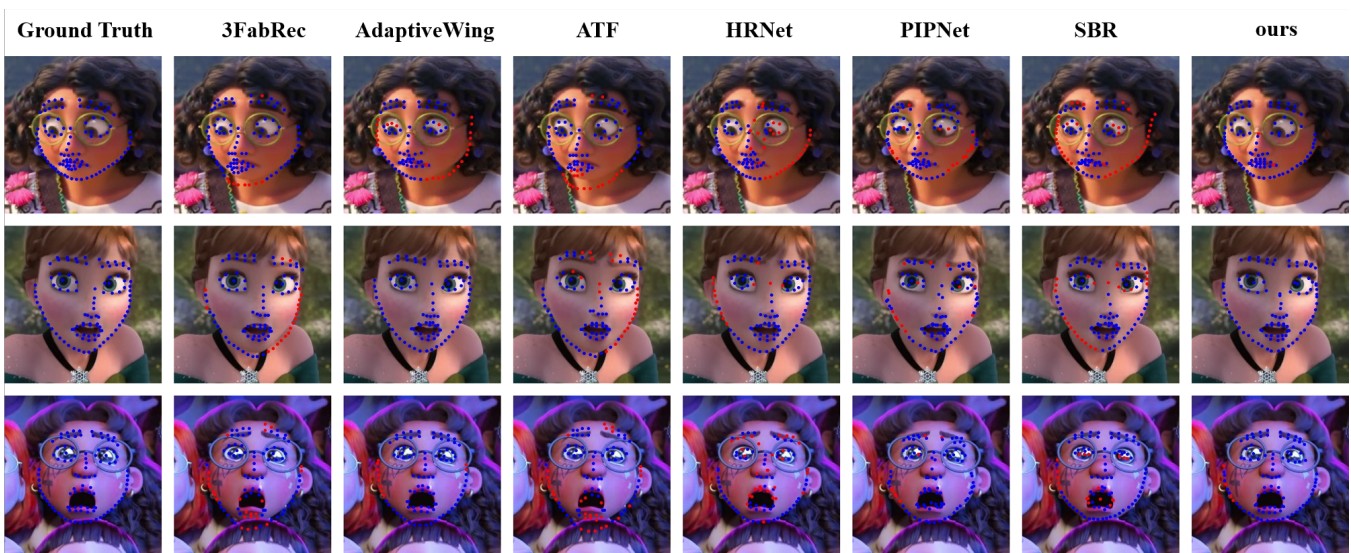

**Figure 5: Comparison between some state-of-the-art landmark detection methods and our method.** *(All models are retained on our FLSC dataset.)* **Images ©Disney Enterprises, Inc.**

| Methods | Backbone | NME (%) | FR$_{10\%}$ | AUC$_{10\%}$ |
|---------|----------|---------|-------------|--------------|
| SBR[10] | VGG-16 | 7.05 | 10.83 | 0.3673 |
| AWing[37] | Hourglass | 7.87 | 14.82 | 0.2968 |
| HRNet[35] | HRNetV2 | 7.41 | 12.20 | 0.3433 |
| ATF[21] | HRNetV2 | 9.20 | 23.29 | 0.2126 |
| 3FabRec[2] | Resnet-18 | 9.04 | 21.54 | 0.2324 |
| PIPNet[17] | Resnet-101 | 5.54 | 7.35 | 0.4835 |
| Ours(retrain) | Hourglass | **5.24** | **6.48** | **0.5313** |
| Ours(fine-tune) | Hourglass | 5.46 | 6.48 | 0.5313 |

**Table 2: Comparision with the state-of-the-art human facial landmark detection models on FLSC. These models are trained on FLSC training set. We export NME score, FR and AUC as evaluation metrics. The thresholds for the latter two are set to 0.1.**

| Methods | Backbone | NME (%) | FR$_{10\%}$ | AUC$_{10\%}$ |
|---------|----------|---------|-------------|--------------|
| dlib[18] | – | 16.32 | 47.45 | 0.1732 |
| Face of Art[41] | FCN | 10.46 | 31.17 | 0.2741 |
| ArtFacePoints[29] | Resnet | 11.93 | 35.62 | 0.2413 |
| Ours(retrain) | Hourglass | **5.24** | **6.48** | **0.5313** |
| Ours(fine-tune) | Hourglass | 5.46 | 6.48 | 0.5313 |

**Table 3: Comparision with the state-of-the-art stylized facial landmark detection models on FLSC. We export NME score, FR and AUC as evaluation metrics. The thresholds for the latter two are set to 0.1.**

*5.4.1 Comparison with models trained on human face dataset.* We conduct a comparison between models trained on the human landmark dataset(WFLW) and our stylized landmark dataset(FLSC). The validation data consists of characters with the style in our dataset

and other styles outside the dataset. The comparative results are presented in Figure 3. Notably, the model that is trained on real human datasets yields unsatisfactory results, while the retrained models exhibit significantly improved accuracy.

Importantly, despite our dataset only consisting of limited styles of characters, the model trained on this dataset demonstrates superior performance when applied to characters with other styles. Therefore, we assert that our dataset can adequately provide models with the features of stylized characters.

*5.4.2 Comparison with existing stylized detectors.* We compare our detector with existing stylized landmark detectors in Figure 4. These landmark detectors produce 68 landmarks. To facilitate comparison, we convert the 98 landmarks generated by our model into 68 landmarks. The result indicates that existing detectors cannot provide accurate results, while the performance of our detector remarkably surpasses current results, especially in the region of the eyes, mouth, and cheek.

*5.4.3 Comparison with existing landmark detection methods.* We compare our method with existing landmarks detection methods in Figure 5. All methods are trained on our stylized landmark dataset. The result shows that our methods do achieve the best result in this task.

Based on these visualizations and the quantitative results above, we believe that our method can perform well enough for stylized landmark detection, and our dataset holds great potential to assist researchers in achieving better results in related tasks. To better illustrate the performance of our detector, we provide a video that contains a 20-second stylized video and its detection result.

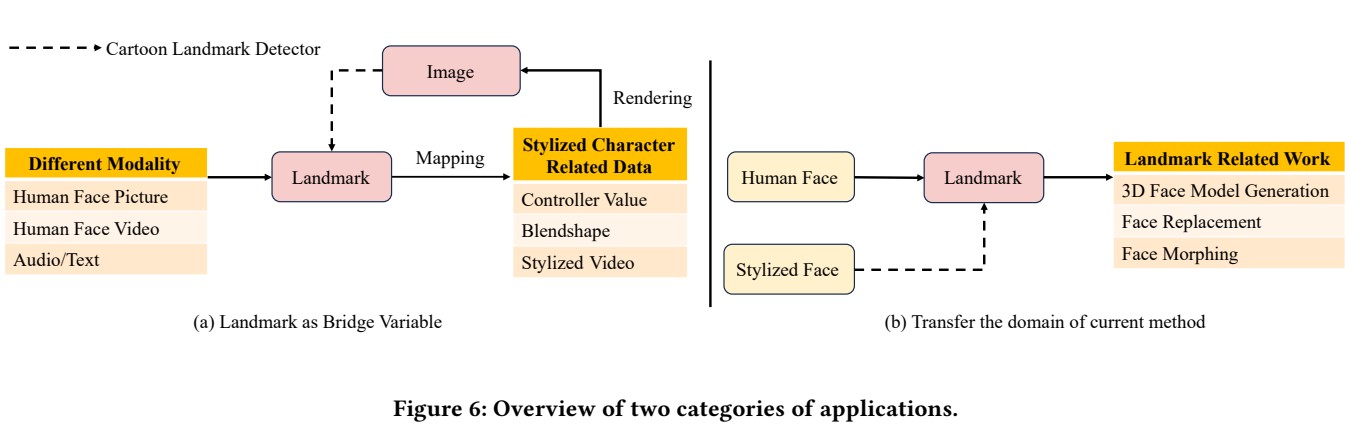

Figure 6: Overview of two categories of applications.

# 6 APPLICATIONS

In this section, we aim to discuss the practical applications of our work. Based on our dataset, the most fundamental application involves training a landmark detector for images featuring stylized characters. Building upon this foundation, we then categorize the subsequent applications into two main groups:

The first category of the application involves leveraging landmarks as a bridge variable. On one end, we have 3D character-related data including cartoon videos and controller values. Through the utilization of the detector trained on our dataset, we can establish a mapping between landmarks and 3D character-related data. On the other end, we encounter diverse data or methods capable of generating landmarks, such as real human faces or results derived from generative models using text or audio. Using the mapping generated before, we can now establish a connection between these 3D character-related data and this landmark-oriented work.

The second category involves transferring tasks that currently rely on landmarks to a new domain. These tasks are typically performed within the context of human faces, where landmarks serve as pivotal intermediary variables. With the aid of our detector, we can extend these tasks to the domain of stylized characters.

Here we will provide two illustrative examples to demonstrate the application of our detector in these two categories.

## 6.1 Bridge Audio and Controller Values

As depicted in Figure 6, landmarks play a crucial role in bridging the gap between stylized characters and other works that involve landmark generation.

*Introduction.* In certain cases, when we require 3D stylized characters to exhibit various facial expressions, we employ controller values to control their facial movements. Unlike other control methods such as blendshapes, this form of control enables direct manipulation of specific facial skeletal structures on the character. However, little research has been conducted on converting 2D stylized landmarks into controller values. Hence, it becomes necessary to utilize our dataset to train a mapping between stylized landmarks and controller values.

The field of audio-driven talking heads has gained considerable attention, with several works employing landmarks as an intermediate output, such as Makeittalk[43] and MEAD[36]. The Makeittalk approach accepts images of a specific character as input and, guided

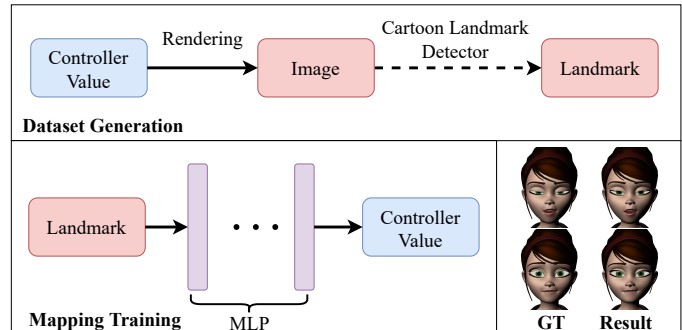

Figure 7: Procedure and results of Audio to Controller Value.

by the audio input, produces a sequence of landmarks depicting the facial movements.

Here, we present an example of how our dataset and method help to train a mapping between landmarks and controller values, further enabling a framework from audio to 3D character animation.

*Method.* The first step in this process involves establishing a mapping between landmarks and controller values. Here we select an open-source character 'Mery', along with a foundational dataset containing its controller values. These controller values represent various expressions and mouth shapes. To construct the mapping dataset, we begin by rendering 2D images using Maya. Subsequently, we employ our fine-tuned model to extract landmarks from these images. The extracted landmarks are then normalized to ensure consistency in scale and position. With a one-to-one corresponding dataset between landmarks and controller values, we can train a Multilayer Perceptron (MLP) to map landmarks to their corresponding controller values.

Next, we need to modify the work Makeittalk[43] to obtain the desired landmarks. The original approach utilizes AdaptiveWingLoss[37] to obtain the initial landmarks. Here we substitute this component with our detector and subject the output landmarks to the same normalization procedure. These normalized landmarks are then fed into the mapping model to generate the controller values. These results are then smoothed and integrated into Maya to animate the character.

*Result.* For the mapping Model, we use MSE and failing rate to measure the accuracy. For those whose MSE loss is larger than 0.06, we define it as a failed sample. For our model, we achieve the following result:

$$MSE: 0.0462, \quad FR: 0.195$$

We present the comprehensive framework and results in Figure 7. It is crucial to note that the original "Make it talk" framework was trained on human talking videos, which introduces inherent inaccuracies and may result in unnatural expressions within the stylized character domain. Hence, we showcase it merely as an example of a potential application of our dataset. Future endeavors, such as retraining the landmark generation model or incorporating transfer learning techniques, hold promise for improving the results and achieving more natural and realistic character expressions.

## 6.2 Transfer Domain of 3D Facial Reconstruction

Currently, many applications are based on the localization of facial landmarks. As depicted in Figure 6, Our landmark detector allows for the extension of these applications from real human faces to stylized faces. Here, we provide an example for better understanding.

*Introduction.* 3D face reconstruction is currently a prominent research field, aiming to transform 2D facial images into 3D meshes. In this context, the availability of facial images and precise facial landmark positions is crucial. Accurate localization of facial landmark positions significantly impacts the effectiveness of 3D reconstruction. However, the absence of a landmark detector specifically designed for stylized characters' faces poses a challenge for the 3D facial reconstruction of stylized characters.

Here, we propose an example of how our dataset and method help to transfer the domain of this task from real humans to stylized characters.

*Method.* A high-performing stylized characters' facial landmark detector can be trained on our FLSC dataset. For a 2D stylized face image, we first determine the approximate bounding box position using the facial landmark positions, followed by cropping and alignment of the facial region. Subsequently, the aligned facial image and its corresponding landmark positions are jointly fed into the 3D reconstruction model. Here we select Deep3DFace[7] as the model for 3D reconstruction.

*Result.* Our model and results are presented in Figure 8. Due to the lack of 3D meshes for stylized characters suitable for training Deep3DFace, we utilized a model trained on human faces. While the reconstructed 3D results exhibit relatively precise details, the stylistic resemblance closely aligns with that of human faces. Future work can build upon this by extending efforts to establish 3D stylized datasets for model training.

## 7 CONCLUSION

Facial landmark detection involves identifying specific points on a face. However, the realm of stylized characters' facial landmarks remains relatively unexplored and presents unique challenges due to differences in facial structures and artistic styles. To address these

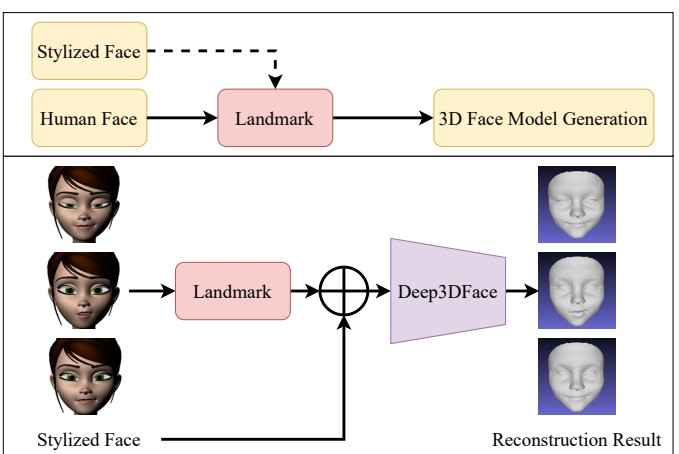

**Figure 8: Procedure and results of 3D facial reconstruction.**

challenges, we introduce the Facial Landmark dataset for Stylized Characters (FLSC).

Based on this dataset, we present StylizedFacePoint, a novel landmark detection framework leveraging a stacked hourglass network. After each stage of the hourglass network, we incorporate offset regression module and neighbor regression module to refine the precision of heatmap regression outcomes. Compared to state-of-the-art methods, StylizedFacePoint demonstrates higher accuracy in landmark detection on stylized face images.

Furthermore, we show potential applications of stylized characters' facial landmark detection. The first category involves using landmarks as bridge variables, and the second category explores domain transfer for tasks reliant on facial landmarks. For each category, we provide one detailed example as an illustration.

In conclusion, our dataset and methods pave the way for face analysis of stylized characters, enabling further landmark-related work in this field.

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
