# OpenReview forum: "StylizedFacePoint: Facial Landmark Detection for Stylized Characters"
_acmmm.org/ACMMM/2024/Conference — MM2024 Poster_

### Official Review · Reviewer_QXYp · 2024-05-24

**Rating:** 3
**Confidence:** 3

**Summary:**

This work tackles the problem of facial landmark detection on 3D stylised characters. In particular, it introduces the first dataset that contains facial landmarks for 3D characters, termed FLSC. It further introduces an end-to-end trainable model, StylizedFacePoint, to conduct landmark detections on stylised characters. Experiments on the introduced FLSC dataset demonstrate the efficacy of the introduced StylizedFacePoint.

**Strengths:**

* It is interesting and meaningful to address the problem of facial landmark detection on 3D stylised characters.
* A 3D stylised character dataset with facial landmarks and facial expressions will benefit the community.

**Limitations:**

* Lack of technical contributions. The introduced StylizedFacePoint appears to be a simple utilisation of the Hourglass Network. It would be better to detail which parts of the model are tailored for stylized characters.

* Lack of comparisons on existing benchmarks. To thoroughly verify the effectiveness of the introduced model, it would be better to also compare it with other existing models on other existing benchmarks, e.g., on the Sindel et al. [29]

* Lack of ablation studies. The overall training objective (Eq. 5) comprises three loss terms. It would be better to conduct ablation studies to verify the main function of each loss term.

* Lack of discussions with more recent work. STAR [ref1] also leverages the hourglass network to achieve facial landmark detection; it sets the state-of-the-art performance, and it is open source. It might be better to compare with STAR to demonstrate the superiority of the introduced StylizedFacePoint


### References:

[ref1] Zhou, Zhenglin, et al. "Star loss: Reducing semantic ambiguity in facial landmark detection." CVPR 2023.

**Suitability:**

2

---

### Official Review · Reviewer_NSJ6 · 2024-05-26

**Rating:** 2
**Confidence:** 2

**Summary:**

StylizedFacePoint is a novel landmark identification framework that uses a stacked hourglass network. Following each level of the hourglass network, we add the offset regression module and the neighbor regression module to improve the precision of heatmap regression results. When detecting landmarks on stylized face photos, StylizedFacePoint outperforms cutting-edge approaches.

**Strengths:**

The proposed study addresses a significant gap in the field of facial landmark detection by introducing the Facial Landmark Dataset for Stylized Characters (FLSC), a comprehensive dataset specifically designed to analyze facial features in stylized characters. This dataset, consisting of a substantial number of images and faces from cartoon video clips, offers a valuable resource for researchers to explore and develop methods tailored to stylized character analysis.

**Limitations:**

One notable limitation pertains to the scalability and diversity of the FLSC dataset. Although the dataset comprises a substantial number of images and faces from various cartoon video clips, its coverage may not fully encapsulate the breadth of stylized character styles encountered in real-world scenarios. Therefore, the generalizability of the proposed method to a broader range of stylized character designs could be further explored and validated through additional data collection efforts or augmentation techniques.

Moreover, while the StylizedFacePoint method demonstrates superior performance, its computational efficiency and robustness to variations in illumination, pose, and facial expression across different character styles require further investigation and optimization. Conducting rigorous experiments and benchmarking against diverse datasets would provide valuable insights into the method's limitations and areas for improvement.

**Suitability:**

2

---

### Official Review · Reviewer_yXZV · 2024-06-04

**Rating:** 4
**Confidence:** 2

**Summary:**

This paper proposes a facial landmark detection model designed for stylized animated characters. When compared to human faces, stylized characters can present unnatural features and proportions which are specific for each animation style, making models trained on real human faces unsuitable for the task.

Their main contributions are their newly proposed dataset (FLSC), which is composed of more than four thousand faces selected from 3D cartoons, with 98 landmarks per face, and a deep learning based method (StylizedFacePoint), trained specifically for landmark detection on stylized animated characters.

The proposed model is a stack of Hourglass Networks that generates several types of heatmaps, from which the x and y landmark positions are extracted. Their main contribution is the combination of a heatmap, offset and neighbour regression losses.

The authors evaluated their StylizedFacePoint method against 6 state-of-the-art methods for human facial landmark detection, and three methods for 2D animated cartoons and mangas, showing that the StylizedFacePoint trained (and not only fine-tune) on the FLSC dataset outperforms the other models.

**Strengths:**

The newly proposed FLSC dataset, which is the first comprising 2D landmarks from 3D animated movies, and its potential applications for 3D reconstruction and generation of facial movements from audio for animated characters.

**Limitations:**

In the section 2.1, the authors present a list of facial landmarks datasets, including real human faces, and 2D cartoon and manga faces in Table 1. However, only the datasets with human faces are described in the text, even though the other cartoon and manga datasets are closer to the task at hand. It is also important to clarify the differences between 2D cartoon, manga and artistic faces.

The authors should make it clear that the proposed dataset contains 2D landmarks. By looking at Table 1, it gives the idea that 3D landmarks were extracted.

A more detailed description of the dataset is missing. What is the proportion of multi-faces, non-frontal and frontal faces? In the test set, what is the proportion of samples obtained from styles not present in the training set? Moreover, in the "Landmarks Model" subsection, the authors should precise that they extracted 2D landmarks.

As mentioned in Section 5.2, the coefficients for the heatmap, offset and neighbour regression losses are fixed to 1, 0.1 and 0.1. How were those values selected?

A few minor suggestions :

Line 111-112: "...train samples..." -> "...training samples..."

Line 305: "Subsequently, We..." -> "Subsequently, we..."

Line 308-310: "Neutral, Sad, Surprise, Happy, Angry, Disgust, and Afraid" -> "Neutral, Sadness, Surprise, Happiness, Disgust and Fear".

Line 336: "normalized heatmap..." -> "normalized heatmaps"

**Suitability:**

2

---

### Meta-Review · Area_Chair_veeC · 2024-07-01

**Recommendation:** Accept (Poster)
**Confidence:** 4

**Metareview:**

This paper proposes a facial landmark detection model designed for stylized animated characters. The main contribution is the proposed dataset (FLSC), which is composed of more than four thousand faces selected from 3D cartoons, and a deep learning based method (StylizedFacePoint), trained specifically for landmark detection on stylized animated characters. The Reviewers did not find an agreement on the paper. Two of them tended toward accepting the paper and found the dataset an interesting contribution. In my views, according to the reviews, the reasons for accepting the paper are prevailing on those for rejecting it. So my suggestion is to have the paper accepted as a poster.